# Identification of a prismatic $P_3N_3$ molecule formed from electron irradiated phosphine-nitrogen ices

Cheng Zhu[1,2], André K. Eckhardt [3✉], Sankhabrata Chandra [1,2], Andrew M. Turner[1,2], Peter R. Schreiner [4] & Ralf I. Kaiser [1,2✉]

Polyhedral nitrogen containing molecules such as prismatic $P_3N_3$ - a hitherto elusive isovalent species of prismane ($C_6H_6$) - have attracted particular attention from the theoretical, physical, and synthetic chemistry communities. Here we report on the preparation of prismatic $P_3N_3$ [1,2,3-triaza-4,5,6-triphosphatetracyclo[2.2.0.0$^{2,6}$.0$^{3,5}$]hexane] by exposing phosphine ($PH_3$) and nitrogen ($N_2$) ice mixtures to energetic electrons. Prismatic $P_3N_3$ was detected in the gas phase and discriminated from its isomers utilizing isomer selective, tunable soft photoionization reflectron time-of-flight mass spectrometry during sublimation of the ices along with an isomer-selective photochemical processing converting prismatic $P_3N_3$ to 1,2,4-triaza-3,5,6-triphosphabicyclo[2.2.0]hexa-2,5-diene ($P_3N_3$). In prismatic $P_3N_3$, the P–P, P–N, and N–N bonds are lengthened compared to those in, e.g., diphosphine ($P_2H_4$), di-anthracene stabilized phosphorus mononitride (PN), and hydrazine ($N_2H_4$), by typically 0.03–0.10 Å. These findings advance our fundamental understanding of the chemical bonding of poly-nitrogen and poly-phosphorus systems and reveal a versatile pathway to produce exotic, ring-strained cage molecules.

[1] Department of Chemistry, University of Hawaii at Manoa, 2545 McCarthy Mall, Honolulu, HI 96822, USA. [2] W. M. Keck Laboratory in Astrochemistry, University of Hawaii at Manoa, 2545 McCarthy Mall, Honolulu, HI 96822, USA. [3] Department of Chemistry, Massachusetts Institute of Technology, Cambridge, MA 02139, USA. [4] Institute of Organic Chemistry, Justus Liebig University, Heinrich-Buff-Ring 17, 35392 Giessen, Germany. ✉email: ake05@mit.edu; ralfk@hawaii.edu

Ever since the postulation of a trigonal prismatic carbon framework for a molecule now known as benzene ($C_6H_6$, **1**) by Albert Ladenburg more than 150 years ago[1], prismanes —a class of hydrocarbons consisting of prism-type polyhedra such as parent prismane ($C_6H_6$, **2**)[2] and cubane ($C_8H_8$, **3**)[3,4]— have fascinated the (in)organic preparative, organometallic, theoretical, and physical chemistry communities from the perspective of fundamental principles of chemical bonding and electronic structure. In conjunction with Langmuir's concept of isovalency, in which molecular entities with the same number of valence electrons have similar chemistries[5], particular attention has been devoted to the preparation of polypnictogen derivatives of prismane, in which the CH moieties are substituted by isovalent nitrogen (N) and phosphorus (P) atoms. These species serve as benchmarks of exotic molecular structures of high strain energies and high-energy-density materials[2,6–12]. From planar, monocyclic $D_{6h}$ symmetric benzene ($C_6H_6$, **1**)[13,14] to polyhedral $D_{3h}$ symmetric prismane (tetracyclo[2.2.0.0$^{2,6}$.0$^{3,5}$]hexane, $C_6H_6$, **2**) (Fig. 1)[2], not only the number of rings increases from one to five, but also the carbon-carbon bond lengths and dihedral angles change from 1.39 Å to 1.52 Å (base edges) and to 1.56 Å (lateral edges) and 0° to 60° (between side face and side face) and 90° (between side face and base), respectively. This induces enhanced energy storage abilities, but also an inherent difficulty in the synthesis and isolation of highly strained valence isomers as authenticated by their distinct thermodynamical stabilities favoring (**1**) by 481 kJ mol$^{-1}$ compared to (**2**) and the time of their first identification of **1** in 1825 versus **2** in 1973.

However, although organometallic molecules with prismatic hexagermanium ($Ge_6$) and hexastannane ($Sn_6$) cores such as 1,2,3,4,5,6-hexakis[bis(trimethylsilyl)methyl]-1,2,3,4,5,6-hex-agerma-tetracyclo[2.2.0.0$^{2,6}$.0$^{3,5}$]hexane [(Me$_3$Si)$_2$CHGe]$_6$[15] and hexasupersilyl-triprismo-hexastannane [(tBu$_3$Si)$_6$Sn$_6$][16,17] have been synthesized, the preparation of molecules isovalent to prismane ($C_6H_6$, **2**), in which all CH groups are replaced by nitrogen and phosphorus atoms, represents a fundamental synthetic challenge due to the repulsive interactions of the lone-pair electrons of phosphorus and nitrogen[18]. Whereas cyclotriphosphazene ($P_3N_3$, **4**) was prepared in low-temperature krypton matrices[6,19] and upon exposing ammonia-phosphine ices to energetic electrons at 5 K[20], prismatic $P_3N_3$ molecules such as 1,2,3-triaza-4,5,6-triphosphatetracyclo [2.2.0.0$^{2,6}$.0$^{3,5}$]hexane ($P_3N_3$, **5**) have remained elusive. Consequently, all-phosphorus-

and nitrogen-substituted prismanes such as **5** represent one of the least explored classes of inorganic molecules.

Here, we show the preparation of prismatic $P_3N_3$ **5**—the isovalent counterpart of prismane **2**—through exposure of low-temperature (5 K) phosphine ($PH_3$) and nitrogen ($N_2$) ice mixtures to ionizing radiation in form of energetic electrons (Methods, Supplementary Tables 1 and 2). By merging our experiments with electronic structure computations, $P_3N_3$ **5** is explicitly identified via vacuum ultraviolet (VUV) photoionization reflectron time-of-flight mass spectrometry (PI-ReTOF-MS) in the temperature-programmed desorption (TPD) phase of the irradiated ices (Supplementary Table 3) based on the computed adiabatic ionization energies (IEs) of distinct $P_3N_3$ isomers (Fig. 2, Supplementary Table 4, Supplementary Data 1) and a mass shift of the ion signal upon $^{15}$N substitution. Isomer-selective photochemical processing at 547 nm converts **5** to 1,2,4-triaza-3,5,6-triphosphabicyclo[2.2.0]hexa-2,5-diene ($P_3N_3$, **16**)— the isovalent analog of Dewar benzene[7]. The identification of **5** offers fundamental insights into pathway(s) towards its synthesis, photochemistry, and molecular structures of highly strained, previously elusive molecular systems thus expanding our knowledge on how we rationalize chemical bonding of nitrogen and phosphorus in polycyclic inorganic molecules.

## Results and discussion

**Infrared spectroscopy.** Fourier-Transform Infrared spectroscopy (FTIR) was utilized to monitor the chemical evolution of the phosphine–nitrogen ices induced by the electron irradiation at 5 K (Supplementary Fig. 1). Nitrogen is infrared inactive; therefore, only absorptions of phosphine were identified in the spectrum of the pristine ice with prominent fundamentals visible at, e.g., 2314 cm$^{-1}$ ($v_3$), 1097 cm$^{-1}$ ($v_4$), and 983 cm$^{-1}$ ($v_2$)[21]. The radiation exposure decomposed 40 ± 5% of the phosphine. This process produced two shoulders at 2270 cm$^{-1}$ and 1063 cm$^{-1}$ and a distinct absorption at 788 cm$^{-1}$; these features can be associated with P−H stretching modes[21], $PH_2$ scissoring modes, and the deformation mode of phosphorus (P) and nitrogen (N) containing rings[22], respectively. The substitution of $N_2$ by $^{15}N_2$ only shifted the 788 cm$^{-1}$ peak to 784 cm$^{-1}$ thus support the aforementioned finding that the structural moiety associate with this absorption has to contain nitrogen. However, since energetic electron irradiation can produce a wide inventory of new species whose absorptions of the functional groups often overlap in the infrared regime[21], infrared spectroscopy is able to determine newly formed functional groups, but does not allow an identification of individual molecules in case of complex mixtures. Therefore, an alternative method is required to probe discrete isomers selectively.

**Mass spectrometry.** To achieve this objective, we took advantage of PI-ReTOF-MS during the TPD phase of the irradiated ices to 300 K[23,24]. This method signifies an exceptional technique of detecting gas-phase molecules isomer-selectively via soft photoionization based on their distinct IEs by systematically tuning the photon energies above and below the IE of the isomer of interest. This ensures the identification of the parent ions at well-defined mass/charge ratio ($m/z$). Considering the computed IEs of individual $P_3N_3$ isomers (Fig. 2), four distinct photon energies (PEs) of 10.49 eV, 8.53 eV, 8.47 eV, and 8.20 eV were selected. Photons at 10.49 eV can ionize all isomers except **6** (IE = 10.97–11.14 eV). Isomer **7** (IE = 10.41–10.58 eV) may or not be ionized by 10.49 eV light. However, this uncertainty will not affect the identification of **5** (IE = 8.36–8.53 eV). As the IEs of **5** (IE = 8.36–8.53 eV) and **16** (IE = 8.51–8.68 eV) overlap in the region of 8.51–8.53 eV, we selected photon energies of 8.53 eV

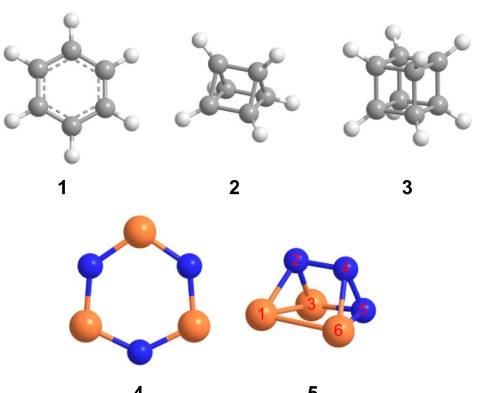

**Fig. 1 Molecular structures of representative $C_6H_6$, $C_8H_8$, and $P_3N_3$ isomers. 1**, benzene ($C_6H_6$, $D_{6h}$); **2**, tetracyclo[2.2.0.0$^{2,6}$.0$^{3,5}$]hexane (prismane, $C_6H_6$, $D_{3h}$); **3**, pentacyclo[4.2.0.0$^{2,5}$.0$^{3,8}$.0$^{4,7}$]octane (cubane, $C_8H_8$, $O_h$); **4**, cyclotriphosphazene ($P_3N_3$, $D_{3h}$); **5**, 1,2,3-triaza-4,5,6-triphosphatetracyclo [2.2.0.0$^{2,6}$.0$^{3,5}$] hexane ($P_3N_3$, $C_1$) (see "Methods: Theoretical"). The atoms are color coded in gray (carbon), white (hydrogen), blue (nitrogen), and orange (phosphorous).

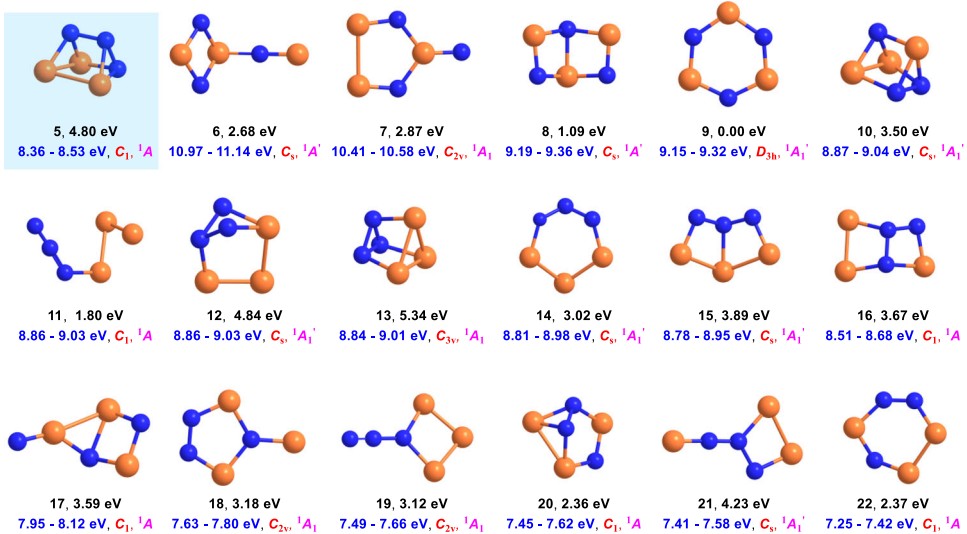

**Fig. 2 Molecular structures of P₃N₃ isomers.** The structures were computed at the B3LYP/cc-pVTZ level of theory. Relative energies in eV (black), computed adiabatic ionization energies corrected for the electric field effect (Stark effect) and computational errors (see Methods: Theoretical) (blue), point groups (red), and ground states (magenta) are also shown. The atoms are color coded in blue (nitrogen) and orange (phosphorous).

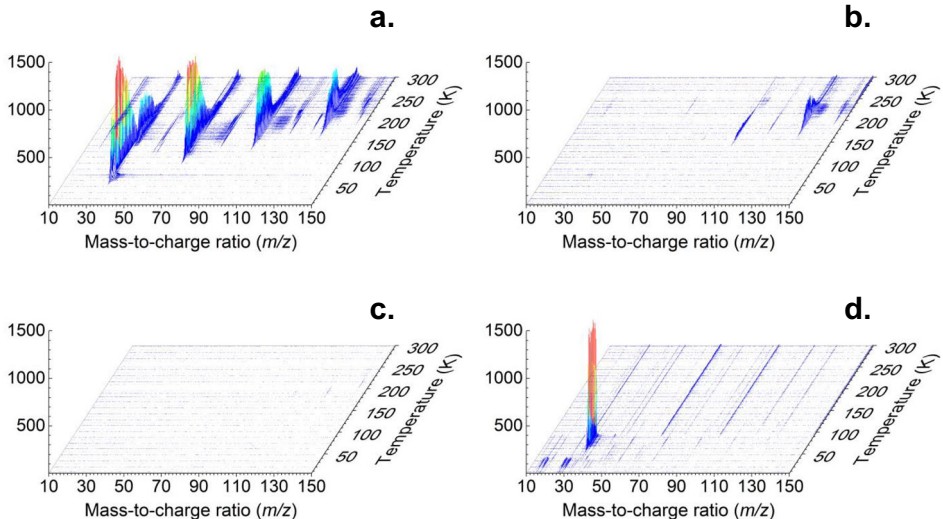

**Fig. 3 Mass spectra showing the temperature-programmed desorption (TPD) profiles for the electron processed phosphine–nitrogen ices.** The data were recorded via photoionization reflectron time-of-flight mass spectrometry (PI-ReTOF-MS) at photon energies of (**a**) 10.49 eV; (**b**), 8.53 eV; (**c**), 8.20 eV; (**d**), blank, 10.49 eV.

and 8.47 eV: 8.53 eV photons can ionize **5** (IE = 8.36–8.53 eV) and isomers **17** (IE = 7.95–8.12 eV) to **22** (IE = 7.25–7.42 eV); **16** (IE = 8.51–8.68 eV) may or may not be ionized with 8.53 eV photons; however, 8.47 eV photons cannot ionize **16** (IE = 8.51–8.68 eV). Finally, 8.20 eV photons are able to ionize **17** (IE = 7.95–8.12 eV) to **22** (IE = 7.25–7.42 eV) if present, but not the target molecule **5** (IE = 8.36–8.53 eV). Overall, a comparison of the TPD traces of the molecular ion counts at $m/z = 135$ (P₃N₃⁺) leads to the identification of **5**. In detail, Fig. 3 compiles the temperature-dependent mass spectra of the desorbed molecules from the electron processed ices upon photoionization in the gas phase. The TPD profiles of the target ions at $m/z = 135$ (P₃N₃⁺) are of particular interest (Fig. 4). At 10.49 eV, the signal at $m/z = 135$ exhibits a maximum at 222 K with a shoulder at about 250 K (Fig. 4a). In a separate experiment replacing nitrogen by 15-nitrogen, both peaks shifted by 3 amu to $m/z = 138$ (Supplementary Fig. 2). These results indicate that the carrier of the $m/z = 135$ contains three nitrogen atoms and hence can only be

assigned to a molecule with the molecular formula P₃N₃. When the photon energy is lowered to 8.53 eV, a main peak at 226 K is present, while the second sublimation event at 250 K vanishes (Fig. 4b). This suggests that the sublimation event at 226 K may be linked to **5** (IE = 8.36–8.53 eV) and/or isomers from **16** (IE = 8.51–8.68 eV) to **22** (IE = 7.25–7.42 eV). The 250 K sublimation event may be connected to isomers from **8** (IE = 9.19–9.36 eV) to **16** (IE = 8.51–8.68 eV). At PE = 8.47 eV, the sublimation event at 226 K is still present and has an identical profile as for 8.53 eV after scaling (Fig. 4b)); therefore, this ion signal can be attributed to **5** (IE = 8.36–8.53 eV) and/or isomers **17** (IE = 7.95–8.12 eV) to **22** (IE = 7.25–7.42 eV), but not to **16** (IE = 8.51–8.68 eV). Tuning down the photon energy to 8.20 eV eliminates the 226 K peak (Fig. 4c). The lack of signal at 8.20 eV suggests that none of the isomers between **17** and **22** formed, and that the ion counts can only originate from isomer **5**. Note that isomers **10** and **13** also have prismatic structures, but their IEs are too close to **11**, **12**, and **14** to be discriminated from each other.

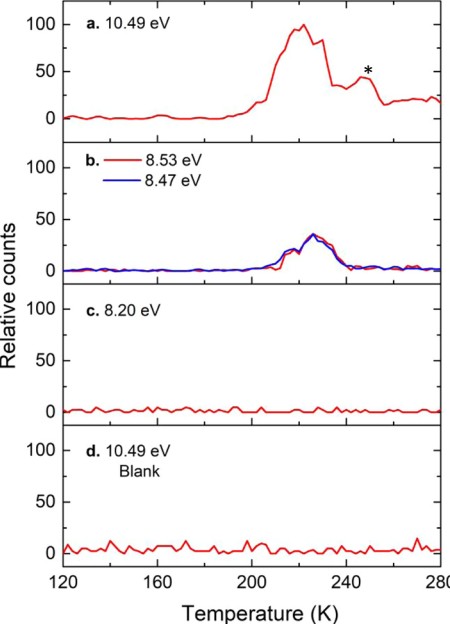

**Fig. 4 PI-ReTOF-MS signals at** $m/z = 135$ **detected during the TPD phase of the electron processed phosphine–nitrogen ices.** The data were recorded at photon energies of (**a**), 10.49 eV; (**b**), 8.53 eV and 8.47 eV; (**c**), 8.20 eV; (**d**), blank, 10.49 eV. The peak marked with an asterisk is not unambiguously assigned.

However, we do not exclude possible contributions of **10** and/or **13** to the sublimation event at 250 K. We stress that we also conducted blank experiments by subliming non-irradiated nitrogen–phosphine ice mixtures at PE = 10.49 eV; no sublimation events were detected at $m/z = 135$ (Fig. 4d) verifying that the identified species are connected with the radiolysis of the ices. Considering the average velocity of 200 m s$^{-1}$ of **5** subliming at 220 K and the 2 mm distance between the ice and the photoionization laser, the lifetimes of the neutral molecule in the gas phase exceeds 10 μs to survive the flight time from the sublimation to the photoionization region.

Since the IEs of **5** and **16** are close to each other, we performed additional experiments to strengthen our identification of **5**. This is achieved through isomer-selective ultraviolet–visible (UV–vis) irradiation experiments, which only photolyze and/or isomerize **5**[20], but not **16**. The experiments first process the ices to prepare the P$_3$N$_3$ isomers followed by wavelength- and hence isomer-selective photoisomerization. Time-dependent density functional theory (DFT) computations disclose that the absorption peak at $\lambda = 547$ nm [highest occupied molecular orbital (HOMO) → lowest unoccupied molecular orbital (LUMO) and HOMO → LUMO + 1] is exclusive to **5**; the remaining absorptions of **5** and **16** overlap (Supplementary Fig. 3, Supplementary Data 2). Therefore, 547 nm light was selected to isomerize and/or photolyze **5**. At PE = 8.47 eV, a comparison of the TPD profiles at $m/z = 135$ of the 'electron plus 547 nm irradiation' system with the 'electron only' irradiation (Fig. 5(a)) reveals that the 226 K peak vanishes; this implies that its carrier can be decomposed and/or isomerized to species with IEs higher than 8.47 eV (**6–16**) by 547 nm photons; consequently, the sublimation event at 226 K must be related to **5**. To untangle the photochemistry of **5**, we also tuned the PE to 8.75 eV in the 'electron plus 547 nm irradiation' experiment and observed a new sublimation events at 220 K. Since this peak is presents at PE = 8.75 eV and disappears at PE = 8.47 eV, it can be assigned to **16** (IE = 8.52–8.68 eV), but not **6–15** (IE > 8.75 eV) or **17–22** (IE < 8.47 eV). These findings

suggest that 547 nm irradiation can isomerize **5–16** and further corroborate our identification of **5**. Note that the energy of a single 547 nm photon (219 kJ mol$^{-1}$) is able to overcome the isomerization barrier between **5** and **16** (108 kJ mol$^{-1}$) (Fig. 5b).

Having provided compelling evidence on the preparation and identification of 1,2,3-triaza-4,5,6-triphosphatetracyclo[2.2.0.0$^{2,6}$.0$^{3,5}$] hexane (P$_3$N$_3$, **5**), we shift our attention now to its computed electronic and geometric structure. The molecule holds a $^1A$ electronic ground state. In contrast to highly symmetric $D_{3h}$ molecule **2**, the prismatic structure in **5** is distorted due to heteroatom substitution, which reduces the point group to $C_1$ (Fig. 1). The computed X–P–X bond angles at the B3LYP/cc-pVTZ level of theory in the bases of the prism are much less than 60° as in an equilateral triangle (∠P(1)–P(3)–N(2) = 52.2°, ∠N(2)–P(1)–P(3) = 52.0°, ∠N(4)–P(5)–N(6) = 48.6°); consequently, the X–N–X angles are more than 60° (∠P(1)–N(2)–P(3) = 75.8°, ∠P(6)–N(5)–N(4) = 68.6°, ∠P(6)–N(4)–N(5) = 62.8°) (Fig. 1). In addition, the mixed heteroatom substitution results in bond lengthening. The N–N bond length is 1.487 Å in the base (N(4)–N(5)) while it is enlarged to 1.528 Å (N(2)–N(4)) in the lateral face, which are longer than the N–N single bond in crystalline hydrazine (N$_2$H$_4$, 1.46 Å)[25]. The same trend is observed for the P–P and P–N bonds. Most of the P–P (P(1)–P(3) = 2.23 Å, P(1)–P(6) = 2.34 Å) and P–N (P(1)–N(2) = 1.82 Å, P(3)–N(2) = 1.81 Å, (P3)–N(5) = 1.82 Å, P(6)–N(4) = 1.84 Å, P(6)–N(5) = 1.76 Å,) bonds were computed to be longer than that in diphosphine (P$_2$H$_4$, P–P = 2.26 Å)[21] and PN containing species (e.g., 1.71 Å in carbene stabilized PN[26], 1.70 Å in di-anthracene stabilized PN[27], and 1.72 Å to 1.73 Å in cyclo-tetraphosphazene[28]), respectively. All these differences cause a distorted and therefore high-energy prismatic structure (Fig. 2). Parent carbon and hydrogen based prismane is 509 kJ mol$^{-1}$ higher in energy than its benzene isomer **1**, which is similar to the energy difference between **5** and **4** (463 kJ mol$^{-1}$). We determined the strain energies of **2** and **5** at the CBS-QB3 level of theory based on the homodesmotic equations depicted in Fig. 6 considering that equal numbers of C−C bonds and [P–P, P–N and N–N bonds] are broken/formed in the educts and products for **2** and **5**, respectively. The strain energy for parent prismane is determined to be 540 kJ mol$^{-1}$, which is in good agreement with the summed strain energy of three cyclobutanes (110 kJ mol$^{-1}$) and two cyclopropanes (115 kJ mol$^{-1}$)[29]. The computed strain energy of **5** (382 kJ mol$^{-1}$) is significantly lower than that of **2** due to the heteroatom substitution. Note that the white phosphorous (P$_4$) allotrope prefers a tetrahedral conformation, while tetrahedrane (CH)$_4$ has never been observed due to the high strain energy. Only derivatives of tetrahedrane have been synthesized and investigated so far[30,31]. Based on a natural bond analysis (NBO), the highest occupied molecular orbital (HOMO) and lowest unoccupied molecular orbital (LUMO) are both six-fold degenerate and represent the six $\sigma_{CC}$ bonds in the two three-membered rings and all $\sigma^*_{CH}$ bonds, respectively. In the case of **5**, the HOMO and HOMO−1 represent $\sigma_{PP}$ in the three- and four-membered rings, respectively. The corresponding antibonding orbitals are represented by the LUMO and LUMO + 1. Hence, we expect that these are the most reactive bonds of **5** and will be broken first during potential chemical reactions or ring openings.

We now discuss possible formation pathways to **5**. Previous experiments on the exposure of pure ices of nitrogen (N$_2$) to energetic electrons at 5 K revealed that the highly reactive acyclic azide radical (N$_3$·) represents the sole polyatomic reaction product[32]. Triphosphane (P$_3$H$_5$) was detected as a dominant product in radiolyzed phosphine (PH$_3$) ices[21]. Here, the azide radical and triphosphane carry the acyclic triatomic nitrogen and phosphorus moieties as present in **5**, respectively. Consequently, the formation of **5** might involve these triatomic molecular building blocks, which may lead to **5** upon interaction with ionizing radiation as demonstrated here.

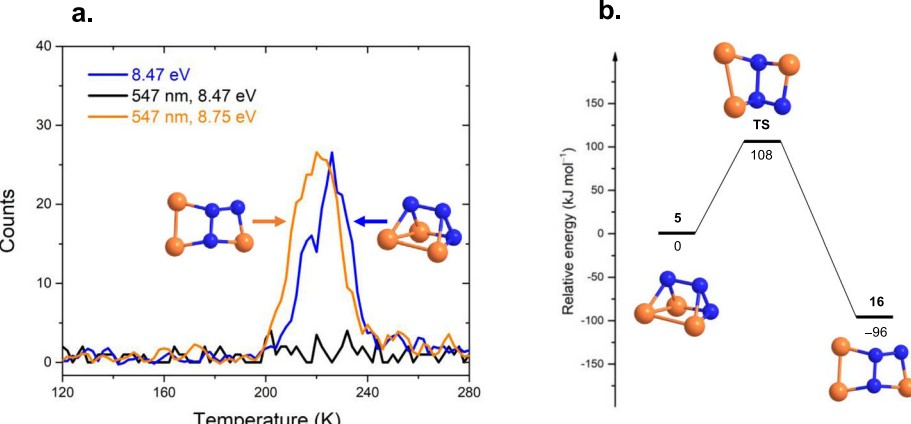

**Fig. 5 Photoisomerization of 1,2,3-Triaza-4,5,6-triphosphatetracyclo [2.2.0.0$^{2,6}$.0$^{3,5}$]hexane (P$_3$N$_3$, 5) by 547 nm laser irradiation. a** PI-ReTOF-MS signal at $m/z = 135$ detected during the TPD phase of the electron and 547 nm laser processed phosphine–nitrogen ice mixture; **b** potential energy surface (PES) for the isomerization of **5**–**16**. The energies were computed at the CCSD(T)/CBS//B3LYP/cc-pVTZ level of theory including zero-point vibrational energy (ZPVE) corrections. The atoms are color coded in blue (nitrogen) and orange (phosphorous).

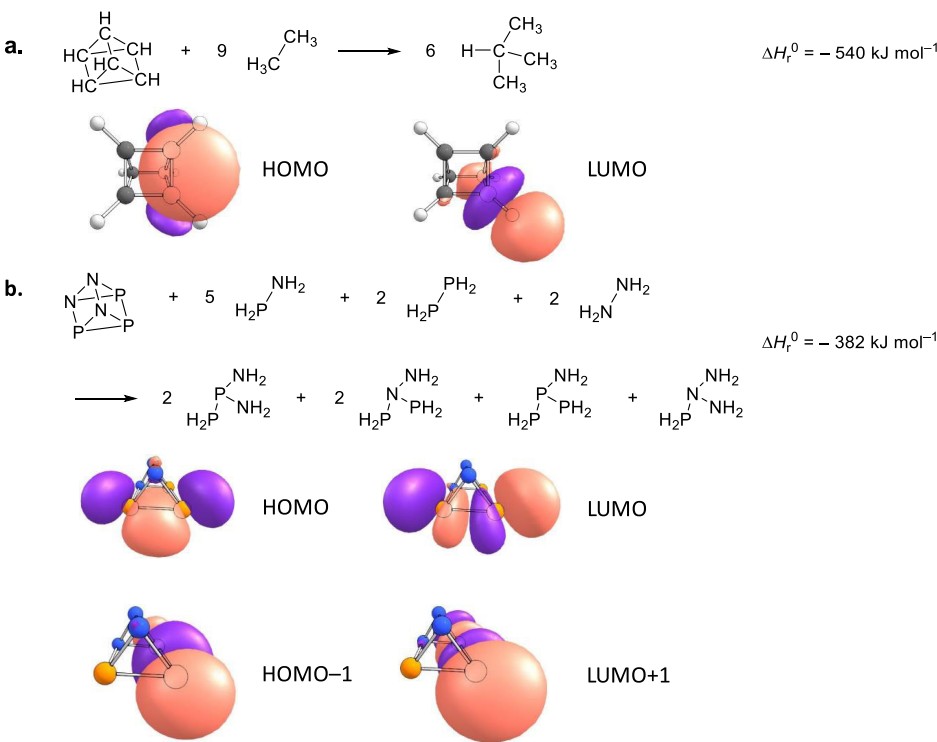

**Fig. 6 Homodesmotic equations for the determination of the strain energies of prismane (C$_6$H$_6$, 2) and prismatic P$_3$N$_3$ (5).** The strain energies of **2** and **5** were computed at the CBS-QB3 level of theory based on the equations in (**a**) and (**b**) considering that equal numbers of C−C bonds and [P−P, P−N and N−N bonds] are broken/formed in the educts and products for **2** and **5**, respectively. Computed frontier natural bond orbitals (NBO), highest occupied molecular orbitals (HOMO), and lowest unoccupied molecular orbitals (LUMO) of **2** and **5** are also shown. Purple and light pink colors correspond to electronically rich and deficient circumstances, respectively. The atoms are color coded in gray (carbon), white (hydrogen), blue (nitrogen), and orange (phosphorous).

Overall, this work not only advances our fundamental perception of the chemical bonding of nitrogen and phosphorus systems, but also provides a potentially viable pathway to generate polyhedral compounds carrying pnictogens such as, for instance, the hitherto elusive hexa-nitrogen (N$_6$) molecules including $D_{6h}$ planar hexaaza benzene[18,33], $C_{2v}$ hexaaza-Dewar benzene[33,34], and $D_{3h}$ hexaaza prismane[35] by extensive electron irradiation of pure nitrogen ice and distinguish them by selectively photoionizing the isomers with VUV photons. This approach may also be

valuable for future studies of other binary P−N compounds similar to P$_3$N$_5$[36], P(N$_3$)$_3$[37], P(N$_3$)$_5$[37], and P$_3$N$_{21}$[38].

## Methods
**Experimental**. The experiments were carried out in an ultra-high vacuum chamber (UHV) pumped to a base pressure of $7 \times 10^{-11}$ Torr utilizing turbomolecular pumps (Osaka, TG1300MUCWB and TG420MCAB) backed with oil-free scroll pumps (Edwards GVSP30)[20,23,39,40]. Within the chamber, a fine polished silver wafer is mounted to a copper cold head cooled by a closed-cycle helium refrigerator (Sumitomo Heavy Industries, RDK-415E) capable of achieving temperatures to

$5.0 \pm 0.1$ K. The wafer can be translated vertically and rotated in the horizontal plane with a linear translator (McAllister, BLT106) and a rotational feedthrough (Thermionics Vacuum Products, RNN-600/FA/MCO), respectively. During the experiment, phosphine ($PH_3$, Sigma-Aldrich, 99.9995%) and nitrogen ($N_2$, Matheson, 99.9992%) (or $^{15}N$-nitrogen ($^{15}N_2$, Sigma-Aldrich, 98% $^{15}N$)) gases (Supplementary Table 1) were co-deposited onto the silver wafer via two separate glass capillary arrays to produce ice mixtures of $PH_3$ and $N_2$ with a composition ratio of (1.2 ± 0.2): 1. The ice thickness was determined exploiting laser interferometry[41] by monitoring the interference fringes of a 632.8 nm helium-neon laser (CVI Melles Griot, 25-LHP-230) that is reflected from the silver wafer (2° relative to the ice surface normal). With the refractive indexes of pure $PH_3$ and $N_2$ ices ($n_{PH_3} = 1.51 \pm 0.04$, $n_{N_2} = 1.21 \pm 0.01$)[21,42] and their composition ratio ($PH_3$ and $N_2 = (1.2 \pm 0.2): 1$), the thickness of the ice mixture was estimated to be $1000 \pm 50$ nm.

After the deposition, the ices were examined utilizing a Fourier transform infrared (FTIR) spectrometer (Nicolet 6700, 6000–400 cm$^{-1}$, 4 cm$^{-1}$ spectral resolution). Supplementary Fig. 1 shows the FTIR spectra of pristine $PH_3 + N_2$ ice. The ice composition was calculated using a modified Beer-Lambert law[21]. For $PH_3$, based on the absorption coefficients for the 2319 cm$^{-1}$ ($\nu_1/\nu_1$, $4.7 \times 10^{-18}$ cm molecule$^{-1}$) and 983 cm$^{-1}$ ($\nu_2$, $5.1 \times 10^{-19}$ cm molecule$^{-1}$) bands[21] along with corresponding integrated areas, its average column density was determined to be $(1.5 \pm 0.3) \times 10^{18}$ molecules cm$^{-2}$, which can be converted to $550 \pm 60$ nm thick ice with the density of $PH_3$ ice (0.9 g cm$^{-3}$)[21]. Considering the thickness of the total ice ($1000 \pm 50$ nm) and $PH_3$ ice ($550 \pm 60$ nm), the thickness of $N_2$ was estimated to be $450 \pm 50$ nm, which corresponds to a column density of $(1.2 \pm 0.3) \times 10^{18}$ molecules cm$^{-2}$ taking into account the densities of $N_2$ ice ($0.94 \pm 0.02$ g cm$^{-3}$)[42]. Therefore, the ratio of $PH_3$ and $N_2$ was found to be $(1.2 \pm 0.2): 1$.

The ices were then isothermally processed by 5 keV electrons (Specs EQ 22-35 electron source) at $5.0 \pm 0.1$ K for two hours at currents of 0 nA (blank) and 100 nA (Supplementary Table 1). The electron incidence angle is 70º to the ice surface normal. Utilizing Monte Carlo simulations (CASINO 2.42)[43], the maximum and average depths of the electrons were estimated to be $880 \pm 90$ and $490 \pm 50$ nm, respectively (Supplementary Table 2), which are less than the ice thickness ($1000 \pm 50$ nm) avoiding interaction between the electrons and the silver wafer. The irradiation doses were determined to be $26 \pm 4$ eV per $PH_3$ molecule and $21 \pm 3$ eV per $N_2$ molecule (Supplementary Table 2). To monitor the evolution of the ices during the electron irradiation, in situ FTIR spectra were recorded at intervals of 2 min.

After the irradiation, the ices were annealed to 300 K at 1 K min$^{-1}$ (temperature-programmed desorption (TPD) exploiting a programmable temperature controller (Lakeshore 336) or first irradiated by 547 nm laser light ($8 \times 10^{21}$ photons) and then the TPD. The 547 nm laser light was generated using the second harmonic (532 nm) of a pulsed neodymium-doped yttrium aluminum garnet laser (Nd:YAG, Spectra Physics, PRO-270, 30 Hz) to pump a Pyrromethene (0.20 g L$^{-1}$ ethanol) dye (0.37 g L$^{-1}$ ethanol). During the TPD phase, the sublimiting molecules from the ices were examined using a reflectron time-of-flight (ReTOF) mass spectrometer (Jordon TOF Products, Inc.) coupled with tunable vacuum ultraviolet (VUV) photon ionization[39] (Figs. 2 to 4, Supplementary Fig. 2). Five photon energies of 10.49 eV, 8.75 eV, 8.53 eV, 8.47 eV, and 8.20 eV were selected to distinguish the $P_3N_3$ isomers. The 10.49 eV (118.222 nm) laser was generated via frequency tripling ($\omega_{vuv} = 3\omega_1$) of the third harmonic (355 nm) of the fundamental (1064 nm) of a Nd:YAG laser using xenon (Xe) as a non-linear medium. To produce 8.75 eV (141.696 nm) light, the second harmonic (532 nm) of a Nd:YAG laser was used to pump a Rhodamine 610/640 dye mixture (0.17/0.04 g L$^{-1}$ ethanol) to obtain 606.948 nm (2.04 eV) (Sirah, Cobra-Stretch), which underwent a frequency tripling process to achieve $\omega_1 = 202.316$ nm (6.13 eV) ($\beta$-BaB$_2$O$_4$ (BBO) crystals, 44° and 77°). A second Nd:YAG laser (second harmonic at 532 nm) pumped LDS 722 dye (0.25 g L$^{-1}$ ethanol) to obtain $\omega_2 = 707$ nm (1.75 eV), which underwent a frequency doubling process to achieve $\omega_2 = 353.5$ nm (3.51 eV)) ($\beta$-BaB$_2$O$_4$ (BBO) crystals, 44°) and then combined with $2\omega_1$ generating $\omega_{vuv} = 141.696$ nm (8.75 eV) at $10^{12}$ photons per pulse via difference four wave mixing ($\omega_{vuv} = 2\omega_1 - \omega_2$) in pulsed gas jets of krypton (Kr). The settings for generating 8.53 eV (145.351 nm), 8.47 eV (146.380 nm), and 8.20 eV (151.200 nm) photons are the same as that for 8.75 eV (141.696 nm) except tuning $\omega_2$ to 332.5 nm (3.73 eV), 327.5 nm (3.78 eV), and 305.5 nm (4.06 eV) via substituting the LDS 722 dye by DCM dye (0.30 g L$^{-1}$ dimethyl sulfoxide), DCM dye (0.30 g L$^{-1}$ dimethyl sulfoxide), and Rhodamine 610/640 dye mixture (0.17/0.04 g L$^{-1}$ ethanol) to generate 665 nm (1.86 eV), 655 nm (1.89 eV), 611 nm (2.03 eV), respectively. The VUV photons were spatially separated from the incident lasers ($2\omega_1$ and $\omega_2$) and other wavelengths generated via multiple resonant and non-resonant processes ($2\omega_1 + \omega_2$; $3\omega_1$; $3\omega_2$)) utilizing a biconvex lens made with lithium fluoride (LiF) (ISP Optics) and directed 2 mm above the ice surface for ionizing the sublimed species. The ionized molecules were examined with the ReTOF mass spectrometer based on their arrival time to a multichannel plate (MCP), which is correlated with mass-to-charge ratios (m/z). The MCP signal was first amplified by a fast preamplifier (Ortec 9305) and then recorded using a multichannel scalar (MCS) (FAST ComTec, P7888-1 E). The MCS is triggered with a pulse generator (Quantum Composers 9518) at 30 Hz. Each final mass spectrum is the average of 3600 sweeps of the flight time in 4 ns bin width and correlates to 2 K increase of the sample temperature.

**Theoretical**. All quantum mechanical calculations were conducted with Gaussian 16, Revision A.03[44] (Fig. 2, Supplementary Table 4, Supplementary Data 1 and 2). Density functional theory (DFT) using the B3LYP functional[45–47] with the

Dunning correlation consistent split valence basis set cc-pVTZ[48] was employed to optimize geometry and calculate frequencies. The frozen-core coupled cluster[49–52] CCSD(T)/cc-pVDZ, CCSD(T)/cc-pVTZ and CCSD(T)/cc-pVQZ single point energies at optimized geometries were calculated and extrapolated to complete basis set limit CCSD(T)/CBS[53] with zero-point vibrational energy (ZPVE) corrections at the B3LYP/cc-pVTZ level of theory. The adiabatic ionization energies (IEs) were determined based on the difference of the ZPVE corrected energies of neutral molecule and corresponding ionic species with similar conformation. The electric field of the extractor plate of our experimental setup lowers the ionization energy by up to 0.03–0.05 eV (Stark effect)[54,55]. The ultraviolet–visible (UV–Vis) spectra for $P_3N_3$ isomers **5** and **16** were computed exploiting time-dependent (TD) B3LYP method using a cc-pVTZ basis set.

## Data availability

All data generated in this study are provided in the article, Supplementary Information, and Supplementary Data files.

## Code availability

The authors did not use any previously unreported custom computer code or algorithm.

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

## Acknowledgements

The experiments at the University of Hawaii were supported by the US National Science Foundation (NSF), Division for Astronomy (NSF-AST 1800975). The W. M. Keck Foundation and the University of Hawaii at Manoa financed the construction of the experimental setup. A.K.E. thanks the Alexander von Humboldt Foundation for a Feodor Lynen research fellowship.

## Author contributions

R.I.K designed experiments. C.Z., S.C. and A.M.T. performed experiments. A.K.E. and P.R.S. carried out the theoretical analysis. C.Z., A.K.E. and R.I.K. wrote the manuscript, which was read, revised, and approved by all co-authors.

## Competing interests

The authors declare no competing interests.
