## [Peer Review File · Nature Communications]

REVIEWER COMMENTS

Reviewer #1 (Remarks to the Author):

This communication is in the finest tradition of Kaiser et al. skillfully interweaving sophisticated experimental techniques with high-level quantum chemical computations and state-of-the-art spectroscopy. Here the authors report the first preparation of prismatic P₃N₃ (5), an analogue of the isovalent prismane, C₆H₆. The authors have to get congratulated on their outstanding work and break-through findings. This communication will be of highest interest to the entire chemistry community, including synthetic, inorganic, organic, physical, theoretical chemistry as well as spectroscopy. The work could not have been carried out more thoroughly and the scholar presentation of this manuscript is excellent.

Also, there are not many neutral binary P-N compounds known, e.g. P₃N₅, P(N₃)₃, P(N₃)₅ and P₃N₂₁. It might be interesting to mention this in the introduction (but I do not make this mandatory).

With respect to future work, the authors may also want to consider the preparation of the still elusive isovalent prismatic N₆.

I recommend accelerated publication without revision.

Reviewer #2 (Remarks to the Author):

In the manuscript entitled "A P₃N₃ Prism: 1,2,3-Triaza-4,5,6-triphosphatetracyclo [2.2.0.0 2,6 .0 3,5]hexane", C. Zhu and coauthors present an experimental work on the formation of P₃N₃ following irradiation by keV electrons of a condensed film made of a mixture of phosphine and nitrogen. Using VUV photoionisation mass spectrometry of sublimated P₃N₃ molecules they can identify isomers. Adiabatic ionisation potentials (AIP), electronic and molecular structures are calculated by density functional theory (DFT) calculations. They claim that a prismatic form of P₃N₃ is formed and identified in the present experiments.

The present manuscript relies on an effective experimental method which previously allowed the identification of numerous elusive species. Molecules formed in the condensed film by irradiation are sublimated by Temperature Programmed Desorption (TPD) and analysed by mass spectrometry after VUV photoionisation. Isotopically labelled molecules are also used to check the mass

spectrometry assignment. The key point of the method is the discrimination of isomer based on VUV photoionisation with photon energies below and above the ionisation potential of a given isomer. Such experimental methods require the knowledge of the different isomers and their respective ionisation potentials which are computed by DFT calculations.

Using the same method and focusing on P3N3 species, the same group recently identified the Dewar isomer of P3N3 cyclotriphosphazene [C. Zhu et al., *Sci. Adv.* 6 (2020) eaba6934; Ref.21 of the manuscript]. The present study differs from the previous one by the mixture used in the condensed film and by a more extensive computational study of the isomers including the prismatic ones. My main concern is the conclusion stating that they identified the prismatic P3N3 (isomer 5) while in my opinion the isomer 16 cannot be excluded.

In the manuscript, the authors calculated the AIP of isomer 16 at 8.61 eV (including a shift due to the extraction field of the time-of-flight spectrometer equal to 0.03 eV). As they consider that the error of their calculations is 0.07 eV and the photon energy is 8.53 eV they conclude that the isomer 16 cannot be ionised. However in a previous work the authors evaluated the ionisation potential lowering due to the Stark effect to be in the 0.03-0.05 eV range [A. Bergantini et al. *Astrophys. J.* 860 (2018) 108, Ref. 52 of the manuscript]. The choice of the value of the lowering will be interesting to discuss. Obviously considering a lowering by 0.05 eV makes possible the ionisation of isomer 16 at a photon energy of 8.53 eV and considering the error of calculations presently evaluated by the authors. This is another matter of concern. The evaluation of such error based on the comparison with only four experimental values (see Suppl. Inf.) seems a bit crude. The vertical ionisation potential may be also interesting to consider.

In conclusion, in my opinion the identification of the prismatic isomer P3N3 does not seem unambiguous. Probably the case of isomer 16 needs to be addressed. Currently it is never discussed/considered. A similar situation appeared in a previous work [Ref. 21], the authors have also two possible isomers of P3N3 after the VUV photoionisation study. They were able to distinguish the Dewar one using isomer selective UV photolysis of the ice before TPD and VUV photoionisation. Finally it will be also interesting to discuss the difference in the formation of different isomers between the two different mixtures: the one used in Ref. 21 (PH3/NH3) and the one in the present work (PH3/N2). Currently few words on the formation mechanisms are given (lines 159-166).

Reviewer #3 (Remarks to the Author):

This is an excellent piece of work of general interest, suitable for Nature Communications. Congratulations! It was very well done and should be published as is. My only comment, the authors should emphasize better that the given structural parameters are computational and not experimentally observed values.

Karl Christe

Reviewer #1

This communication is in the finest tradition of Kaiser et al. skillfully interweaving sophisticated experimental techniques with high-level quantum chemical computations and state-of-the-art spectroscopy. Here the authors report the first preparation of prismatic P_3N_3 (5), an analogue of the isovalent prismane, C_6H_6 . The authors have to get congratulated on their outstanding work and break-through findings. This communication will be of highest interest to the entire chemistry community, including synthetic, inorganic, organic, physical, theoretical chemistry as well as spectroscopy. The work could not have been carried out more thoroughly and the scholar presentation of this manuscript is excellent.

RESPONSE: Thank you.

Also, there are not many neutral binary P-N compounds known, e.g. P_3N_5 , $P(N_3)_3$, $P(N_3)_5$ and P_3N_{21} . It might be interesting to mention this in the introduction (but I do not make this mandatory).

RESPONSE: Since the introduction focuses on prismane and its isovalent species, we added a sentence about P_3N_5 , $P(N_3)_3$, $P(N_3)_5$, and P_3N_{21} in the conclusion (Page 8).

“This approach may also be valuable for future studies of other binary P–N compounds similar to $P_3N_5^1$, $P(N_3)_3^2$, $P(N_3)_5^2$, and $P_3N_{21}^3$.”

With respect to future work, the authors may also want to consider the preparation of the still elusive isovalent prismatic N_6 . I recommend accelerated publication without revision.

RESPONSE: Thank you.

Reviewer #2

In the manuscript entitled “A P_3N_3 Prism: 1,2,3-Triaza-4,5,6-triphosphatetracyclo [2.2.0.0^{2,6}.0^{3,5}]hexane”, C. Zhu and coauthors present an experimental work on the formation of P_3N_3 following irradiation by keV electrons of a condensed film made of a mixture of phosphine and nitrogen. Using VUV photoionisation mass spectrometry of sublimated P_3N_3 molecules they can identify isomers. Adiabatic ionisation potentials (AIP), electronic and molecular structures are calculated by density functional theory (DFT) calculations. They claim that a prismatic form of P_3N_3 is formed and identified in the present experiments.

The present manuscript relies on an effective experimental method which previously allowed the identification of numerous elusive species. Molecules formed in the condensed film by irradiation are sublimated by Temperature Programmed Desorption (TPD) and analysed by mass spectrometry after VUV photoionisation. Isotopically labelled molecules are also used to check the mass spectrometry assignment. The key point of the method is the discrimination of isomer based on VUV photoionisation with photon energies below and above the ionisation potential of a given isomer. Such experimental methods require the knowledge of the different isomers and their respective ionisation potentials which are computed by DFT calculations.

Using the same method and focusing on P_3N_3 species, the same group recently identified the Dewar isomer of P_3N_3 cyclotriphosphazene [C. Zhu et al., Sci. Adv. 6 (2020) eaba6934; Ref.21 of the manuscript]. The present study differs from the previous one by the mixture used in the condensed film and by a more extensive computational study of the isomers including the prismatic ones. My main concern is the conclusion stating that they identified the prismatic P_3N_3 (isomer 5) while in my opinion the isomer 16 cannot be excluded.

RESPONSE: We performed additional experiments to exclude **16** and to confirm the assignment of **5**. Please see the detailed response below.

In the manuscript, the authors calculated the AIP of isomer 16 at 8.61 eV (including a shift due to the extraction field of the time-of-flight spectrometer equal to 0.03 eV). As they consider that the

error of their calculations is 0.07 eV and the photon energy is 8.53 eV they conclude that the isomer 16 cannot be ionised. However in a previous work the authors evaluated the ionisation potential lowering due to the Stark effect to be in the 0.03-0.05 eV range [A. Bergantini et al. *Astrophys. J.* 860 (2018) 108, Ref. 52 of the manuscript]. The choice of the value of the lowering will be interesting to discuss. Obviously considering a lowering by 0.05 eV makes possible the ionisation of isomer 16 at a photon energy of 8.53 eV and considering the error of calculations presently evaluated by the authors. This is another matter of concern. The evaluation of such error based on the comparison with only four experimental values (see Suppl. Inf.) seems a bit crude.

RESPONSE: The Supplementary Table 5 has been expanded. Considering the Stark effect of -0.03 to -0.05 eV, the IEs of **5** and **16** are in the ranges of 8.36 - 8.53 eV and 8.51 - 8.68 eV, respectively. Photons with energy of 8.53 eV may ionize **16**; therefore we performed an additional experiment at a photon energy (PE) = 8.47 eV, which cannot ionize **16**; we still observed the sublimation events at 226 K with the same TPD profile as that at PE = 8.53 eV after scaling, revealing that this originates from molecule **5**. We also performed additional *isomer selective UV photoisomerization experiments*. Details are described in the response to the next comment of the referee. These findings reveal that the 226 K peak can only originate from **5**. To account for this comment, we modified paragraph 5 (**Mass spectrometry**, Page 5), Fig. 1, Fig. 3, and Supplementary Table 5. The revised text is highlighted in the main manuscript.

In conclusion, in my opinion the identification of the prismatic isomer P_3N_3 does not seem unambiguous. Probably the case of isomer 16 needs to be addressed. Currently it is never discussed/considered. A similar situation appeared in a previous work [Ref. 21], the authors have also two possible isomers of P_3N_3 after the VUV photoionisation study. They were able to distinguish the Dewar one using isomer selective UV photolysis of the ice before TPD and VUV photoionisation.

RESPONSE: As mentioned above, we performed an additional experiment at PE= 8.47 eV. We also computed the UV-vis spectra of **5** and **16**. The results revealed that absorption band at $\lambda = 547$ nm [highest occupied molecular orbital (HOMO) \rightarrow lowest unoccupied molecular orbital (LUMO) and HOMO \rightarrow LUMO + 1] is exclusive to **5** and other absorptions are overlapped (new

Supplementary Figure 3 and Supplementary Table 6). Therefore, we carried out irradiation experiments at $\lambda = 547$ nm to selectively isomerize and/or photolyze isomer **5**. At PE = 8.47 eV, a comparison of the TPD profiles at $m/z = 135$ of the electron plus 547 nm irradiation system with the only electron irradiation sample (Fig. 4(a)) shows that the 226 K peak vanishes, which implies that its carrier was decomposed and/or isomerized to other species with IEs higher than 8.47 eV (from **6** to **16**) by 547 nm photons and therefore can only be **5**. To untangle the fate of **5** after the irradiation, we tuned PE to 8.75 eV and observed a new sublimation event at 220 K. Since this peak presents at PE = 8.75 eV and disappears at PE = 8.47 eV, it is assigned to **16** (IE = 8.52 to 8.68 eV) but not isomers from **6** to **15** (IE > 8.75 eV) or from **17** to **22** (IE < 8.47 eV). These findings suggest that 547 nm irradiation can isomerize **5** to **16** and further corroborate our identification of **5**. Note that the energy of 547 nm photon (219 kJ mol^{-1}) is able to overcome the isomerization barrier between **5** and **16** (108 kJ mol^{-1}) (Fig. 4(b)). To account for this comment, we added the following paragraph in Page 6, Fig. 4, Supplementary Figure 3, and Supplementary Table 6.

“Since the IEs of **5** and **16** are close to each other, we performed additional experiments to strengthen our identification of **5**. This is achieved through isomer-selective ultraviolet-visible (UV-vis) irradiation experiments, which *only* photolyze and/or isomerize **5**⁴, but *not* **16**. The experiments first process the ices to prepare the P_3N_3 isomers followed by wavelength- and hence isomer-selective photoisomerization. Time-dependent density functional theory (DFT) computations disclose that the absorption peak at $\lambda = 547$ nm [highest occupied molecular orbital (HOMO) \rightarrow lowest unoccupied molecular orbital (LUMO) and HOMO \rightarrow LUMO + 1] is exclusive to **5**; the remaining absorptions of **5** and **16** overlap (Supplementary Figure 3, Supplementary Table 6). Therefore, 547 nm light was selected to isomerize and/or photolyze **5**. At PE = 8.47 eV, a comparison of the TPD profiles at $m/z = 135$ of the ‘electron plus 547 nm irradiation’ system with the ‘electron only’ irradiation (Fig. 4(a)) reveals that the 226 K peak vanishes; this implies that its carrier can be decomposed and/or isomerized to species with IEs higher than 8.47 eV (**6** to **16**) by 547 nm photons; consequently, the sublimation event at 226 K must be related to **5**. To untangle the photochemistry of **5**, we also tuned the PE to 8.75 eV in the ‘electron plus 547 nm irradiation’ experiment and observed a new sublimation events at 220 K. Since this peak is presents at PE = 8.75 eV and disappears at PE = 8.47 eV, it can be assigned to

16 (IE = 8.52 to 8.68 eV), but not **6** to **15** (IE > 8.75 eV) or **17** to **22** (IE < 8.47 eV). These findings suggest that 547 nm irradiation can isomerize **5** to **16** and further corroborate our identification of **5**. Note that the energy of a single 547 nm photon (219 kJ mol^{-1}) is able to overcome the isomerization barrier between **5** and **16** (108 kJ mol^{-1}) (Fig. 4(b)).”

“

Fig. 4 Photoisomerization of 1,2,3-Triaza-4,5,6-triphosphatetracyclo [2.2.0.0^{2,6}.0^{3,5}]hexane (P₃N₃, **5) by 547 nm laser irradiation. a, PI-ReTOF-MS signal at $m/z = 135$ detected during the temperature programmed desorption (TPD) phase of the electron and 547 nm laser processed phosphine – nitrogen ice mixture; b, potential energy surface (PES) for the isomerization of **5** to **16**.”**

“

Supplementary Figure 3. Computed ultraviolet–visible (UV-Vis) spectra for P₃N₃ isomers **5 and **16** at the TD-B3LYP/cc-pVTZ level of theory.”**

“Supplementary Table 6. Computed ultraviolet–visible (UV-Vis) absorptions and assignments for P₃N₃ isomers **5** and **16** at the TD-B3LYP/cc-pVTZ level of theory.

Isomer	Wavelength (nm)	Oscillator strength (f)	Transition assignment	Orbital transitions
5	547.24	0.0001	HOMO → LUMO HOMO → LUMO+1	33 → 34 (0.67513) 33 → 35 (0.20098)
	453.27	0.0029	HOMO-1 → LUMO HOMO → LUMO HOMO → LUMO+1	32 → 34 (0.15854) 33 → 34 (-0.18537) 33 → 35 (0.65773)
	354.65	0.0097	HOMO-1 → LUMO HOMO-1 → LUMO+1 HOMO → LUMO+2	32 → 34 (0.54255) 32 → 35 (-0.32636) 33 → 36 (0.27551)
	348.35	0.005	HOMO-2 → LUMO HOMO-1 → LUMO+1 HOMO → LUMO+2	31 → 34 (0.10225) 32 → 35 (0.49046) 33 → 36 (0.48561)
	335.52	0.0168	HOMO-1 → LUMO HOMO-1 → LUMO+1 HOMO → LUMO+1 HOMO → LUMO+2	32 → 34 (-0.39539) 32 → 35 (-0.37683) 33 → 35 (0.12166) 33 → 36 (0.40396)
	318.67	0.0044	HOMO-3 → LUMO HOMO-2 → LUMO HOMO-2 → LUMO+1	30 → 34 (0.36579) 31 → 34 (0.52406) 31 → 35 (0.26908)
	313.85	0.0008	HOMO-3 → LUMO HOMO-2 → LUMO HOMO-2 → LUMO+1	30 → 34 (0.59793) 31 → 34 (-0.30631) 31 → 35 (-0.18929)
	299.72	0.0087	HOMO-3 → LUMO+1 HOMO-2 → LUMO HOMO-2 → LUMO+1	30 → 35 (0.16961) 31 → 34 (-0.28054) 31 → 35 (0.60383)
	284.71	0.0057	HOMO-3 → LUMO+1 HOMO-2 → LUMO HOMO-2 → LUMO+1	30 → 35 (0.67005) 31 → 34 (0.10305) 31 → 35 (-0.12315)

	280.16	0.0008	HOMO → LUMO+3	33 → 37 (0.69097)
	264.47	0.0143	HOMO-1 → LUMO+2	32 → 36 (0.68037)
	258.8	0.0057	HOMO-4 → LUMO HOMO-1 → LUMO+2 HOMO → LUMO+4	29 → 34 (0.67486) 32 → 36 (-0.10204) 33 → 38 (0.10256)
	246.96	0.0025	HOMO-4 → LUMO+1 HOMO-2 → LUMO+2 HOMO → LUMO+4	29 → 35 (0.60286) 31 → 36 (-0.20406) 33 → 38 (-0.27935)
	241.19	0.0023	HOMO-4 → LUMO+1 HOMO-3 → LUMO+2 HOMO-2 → LUMO+2	29 → 35 (0.17796) 30 → 36 (0.44518) 31 → 36 (0.48872)
	235.66	0.0456	HOMO-5 → LUMO+1 HOMO-4 → LUMO+1 HOMO-3 → LUMO+2 HOMO-2 → LUMO+2 HOMO → LUMO+4	28 → 35 (-0.13790) 29 → 35 (0.23814) 30 → 36 (0.18438) 31 → 36 (-0.16459) 33 → 38 (0.55684)
	231.61	0.0258	HOMO-4 → LUMO+1 HOMO-3 → LUMO+2 HOMO-2 → LUMO+2 HOMO-1 → LUMO+3 HOMO → LUMO+4	29 → 35 (-0.16098) 30 → 36 (0.49758) 31 → 36 (-0.38880) 32 → 37 (0.11653) 33 → 38 (-0.15020)
16	448.88	0.0099	HOMO-2 → LUMO HOMO → LUMO	31 → 34 (0.12564) 33 → 34 (0.68460)
	348.53	0.0085	HOMO-1 → LUMO HOMO → LUMO+1 HOMO → LUMO+2	32 → 34 (-0.42401) 33 → 35 (0.52992) 33 → 36 (0.15625)
	329.61	0.0182	HOMO-2 → LUMO HOMO-1 → LUMO HOMO → LUMO+1 HOMO → LUMO+2	31 → 34 (0.29202) 32 → 34 (0.32504) 33 → 35 (0.12594) 33 → 36 (0.52937)
	318.32	0.0113	HOMO-2 → LUMO	31 → 34 (0.53945)

			HOMO-1 → LUMO+1 HOMO → LUMO+1 HOMO → LUMO+2	32 → 35 (0.14128) 33 → 35 (0.15648) 33 → 36 (-0.37721)
	311.74	0.0315	HOMO-3 → LUMO HOMO-2 → LUMO HOMO-2 → LUMO+1 HOMO-1 → LUMO HOMO → LUMO+1 HOMO → LUMO+2 HOMO → LUMO+3	30 → 34 (-0.13701) 31 → 34 (-0.26824) 31 → 35 (0.15583) 32 → 34 (0.35850) 33 → 35 (0.34127) 33 → 36 (-0.19004) 33 → 37 (0.27530)
	287.01	0.0076	HOMO-3 → LUMO HOMO-1 → LUMO+1 HOMO-1 → LUMO+2 HOMO → LUMO+3	30 → 34 (0.10381) 32 → 35 (-0.22825) 32 → 36 (-0.39788) 33 → 37 (0.48126)
	277.1	0.0244	HOMO-4 → LUMO HOMO-3 → LUMO HOMO-2 → LUMO+1 HOMO-1 → LUMO+1 HOMO-1 → LUMO+2 HOMO → LUMO+3	29 → 34 (0.26696) 30 → 34 (0.10573) 31 → 35 (-0.29945) 32 → 35 (0.41916) 32 → 36 (0.16390) 33 → 37 (0.30025)
	270.89	0.0027	HOMO-4 → LUMO HOMO-3 → LUMO HOMO-2 → LUMO+1 HOMO-1 → LUMO+2 HOMO → LUMO+3	29 → 34 (0.45440) 30 → 34 (0.30012) 31 → 35 (0.12364) 32 → 36 (-0.34400) 33 → 37 (-0.19976)
	267.43	0.0194	HOMO-4 → LUMO HOMO-3 → LUMO HOMO-2 → LUMO+1 HOMO-1 → LUMO+1 HOMO-1 → LUMO+2 HOMO → LUMO+1	29 → 34 (-0.38890) 30 → 34 (0.11891) 31 → 35 (0.10294) 32 → 35 (0.41827) 32 → 36 (-0.29071) 33 → 35 (-0.10224)

264.06	0.0291	HOMO-4 → LUMO HOMO-3 → LUMO HOMO-2 → LUMO+1 HOMO-1 → LUMO+1 HOMO-1 → LUMO+2 HOMO → LUMO+3	29 → 34 (-0.12623) 30 → 34 (0.51585) 31 → 35 (0.23859) 32 → 35 (-0.11202) 32 → 36 (0.30735) 33 → 37 (0.11881)
253.28	0.0102	HOMO-5 → LUMO HOMO-2 → LUMO+2	28 → 34 (0.68554) 31 → 36 (-0.12773)
251.1	0.0026	HOMO-2 → LUMO+1 HOMO-2 → LUMO+2 HOMO-1 → LUMO+3 HOMO → LUMO+4	31 → 35 (0.11439) 31 → 36 (0.12504) 32 → 37 (0.62672) 33 → 38 (-0.23864)
245.93	0.0264	HOMO-3 → LUMO HOMO-3 → LUMO+1 HOMO-2 → LUMO+1 HOMO-2 → LUMO+2 HOMO → LUMO+3 HOMO → LUMO+4	30 → 34 (-0.12574) 30 → 35 (-0.11967) 31 → 35 (0.37778) 31 → 36 (0.41539) 33 → 37 (0.11256) 33 → 38 (0.29076)
236.96	0.0078	HOMO-4 → LUMO+1 HOMO-2 → LUMO+2 HOMO-2 → LUMO+3 HOMO-1 → LUMO+1 HOMO-1 → LUMO+3 HOMO → LUMO+4	29 → 35 (0.11023) 31 → 36 (-0.26147) 31 → 37 (0.12173) 32 → 35 (0.10512) 32 → 37 (0.28448) 33 → 38 (0.52498)
235.43	0.0568	HOMO-4 → LUMO HOMO-4 → LUMO+1 HOMO-3 → LUMO HOMO-3 → LUMO+2 HOMO-2 → LUMO+1 HOMO-2 → LUMO+2 HOMO-2 → LUMO+3	29 → 34 (-0.14929) 29 → 35 (-0.15352) 30 → 34 (0.13138) 30 → 36 (-0.14441) 31 → 35 (-0.30914) 31 → 36 (0.41152) 31 → 37 (0.10817)

			HOMO-1 → LUMO	32 → 34 (0.12432)
			HOMO-1 → LUMO+1	32 → 35 (-0.11598)
			HOMO-1 → LUMO+4	32 → 38 (0.12333)
			HOMO → LUMO+1	33 → 35 (0.13918)
			HOMO → LUMO+4	33 → 38 (0.14022)

Finally it will be also interesting to discuss the difference in the formation of different isomers between the two different mixtures: the one used in Ref. 21 (PH₃/NH₃) and the one in the present work (PH₃/N₂). Currently few words on the formation mechanisms are given (lines 159-166).

RESPONSE: Different nitrogen-containing precursors for the preparation of isomer **5** and cyclotriphosphazene (P₃N₃) (Ref. 21) were selected for the following background. As mentioned in the manuscript, previous experiments revealed that the highly reactive acyclic azide radical (N₃[•]) can be produced by electron irradiation of pure nitrogen (N₂) ices. This is a key moiety in the isomer **5**. For cyclotriphosphazene (P₃N₃), the P and N atoms are alternately distributed. Therefore, single nitrogen-containing molecule NH₃ could be a better precursor. However, the detailed formation mechanisms of these P₃N₃ species cannot be untangled currently and are beyond the scope of this study since these are condensed phase experiments, but not crossed molecular beam studies.

Reviewer #3:

This is an excellent piece of work of general interest, suitable for Nature Communications. Congratulations! It was very well done and should be published as is. My only comment, the authors should emphasize better that the given structural parameters are computational and not experimentally observed values.

Karl Christe

RESPONSE: Thank you. To account for this comment, we modified the first sentence of paragraph 7 (Page 7).

“Having provided compelling evidence on the first preparation and identification of 1,2,3-triaza-4,5,6-triphosphatetracyclo[2.2.0.0^{2,6}.0^{3,5}]hexane (P₃N₃, **5**), we shift our attention now to its **computed** electronic and geometric structure.”

References

- 1 Schnick, W., Lücke, J. & Krumeich, F. Phosphorus nitride P₃N₅: Synthesis, spectroscopic, and electron microscopic investigations. *Chem. Mater.* **8**, 281-286 (1996).
- 2 Buder, W. & Schmidt, A. Phosphorazide und deren Schwingungsspektren. *Z. Anorg. Allg. Chem.* **415**, 263-267 (1975).
- 3 Göbel, M., Karaghiosoff, K. & Klapötke, T. M. The first structural characterization of a binary P–N molecule: The highly energetic compound P₃N₂₁. *Angew. Chem. Int. Ed.* **45**, 6037-6040 (2006).
- 4 Zhu, C. *et al.* The elusive cyclotriphosphazene molecule and its Dewar benzene–type valence isomer (P₃N₃). *Sci. Adv.* **6**, eaba6934 (2020).

REVIEWERS' COMMENTS

Reviewer #2 (Remarks to the Author):

The additional experiment performed at a photon energy of 8.47 eV allows to non-ambiguously identify the prismatic isomer of P3N3.

The other experiment performed on 547 nm photon absorption by the irradiated ices is also very interesting and further evidence the formation of the prismatic isomer.

This was my main concern and it is now clearly addressed with the additional experiments.

Such experiments are a bit frustrating concerning the formation mechanisms but as pointed out by the authors in their answer such condensed phase experiments are difficult to model.

Reviewer #2

The additional experiment performed at a photon energy of 8.47 eV allows to non-ambiguously identify the prismatic isomer of P_3N_3 .

The other experiment performed on 547 nm photon absorption by the irradiated ices is also very interesting and further evidence the formation of the prismatic isomer.

This was my main concern and it is now clearly addressed with the additional experiments.

Such experiments are a bit frustrating concerning the formation mechanisms but as pointed out by the authors in their answer such condensed phase experiments are difficult to model.

RESPONSE: Thank you.